# Shape Classification Using a Single Seal-Whisker-Style Sensor Based on the Neural Network Method

**DOI:** 10.3390/s24165418

**Published:** 2024-08-21

**Authors:** Yitian Mao, Yingxue Lv, Yaohong Wang, Dekui Yuan, Luyao Liu, Ziyu Song, Chunning Ji

**Affiliations:** 1Department of Mechanics, School of Mechanical Engineering, Tianjin University, Tianjin 300072, China; maoyitian@tju.edu.cn (Y.M.); 1018201020@tju.edu.cn (L.L.); 2CCCC First Harbor Engineering Company Ltd. (Key Laboratory of Coastal Engineering Hydrodynamics, CCCC), Tianjin 300461, China; lvyingxue@ccccltd.cn; 3Center for Applied Mathematics and KL-AAGDM, Tianjin University, Tianjin 300072, China; 4State Key Laboratory of Hydraulic Engineering Intelligent Construction and Operation, Tianjin University, Tianjin 300072, China; 2023205351@tju.edu.cn (Z.S.); cnji@tju.edu.cn (C.J.)

**Keywords:** biomimetics, harbor seal whiskers, convolutional neural network, underwater detection, shape classification

## Abstract

Seals, sea lions, and other aquatic animals rely on their whiskers to identify and track underwater targets, offering valuable inspiration for the development of low-power, portable, and environmentally friendly sensors. Here, we design a single seal-whisker-like cylinder and conduct experiments to measure the forces acting on it with nine different upstream targets. Using sample sets constructed from these force signals, a convolutional neural network (CNN) is trained and tested. The results demonstrate that combining the seal-whisker-style sensor with a CNN enables the identification of objects in the water in most cases, although there may be some confusion for certain targets. Increasing the length of the signal samples can enhance the results but may not eliminate these confusions. Our study reveals that high frequencies (greater than 5 Hz) are irrelevant in our model. Lift signals present more distinct and distinguishable features than drag signals, serving as the primary basis for the model to differentiate between various targets. Fourier analysis indicates that the model’s efficacy in recognizing different targets relies heavily on the discrepancies in the spectral features of the lift signals.

## 1. Introduction

Underwater detection technology has a wide range of applications in marine resource development, environmental protection, military reconnaissance, and more. Common underwater detection methods include sonar, based on ultrasound, and vision, based on optics. However, both techniques require a large, heavy platform and lead to substantial energy consumption [1,2] Furthermore, sonar detectors may harm aquatic animals that rely on sonar positioning, such as whales [3]. Optical-based vision methods may require additional lighting devices to be used in dark and turbid waters [4]. On the other hand, many marine animals have the natural ability to perceive their surroundings and environment by sensing water flow disturbances through their whiskers, particularly for low-frequency signals. Inspired by this, researchers have endeavored to mimic marine organisms’ whiskers and to develop sensors that passively detect changes in water flow to perceive their surroundings. Compared with traditional sonar systems, these sensors offer advantages such as low energy consumption, minimal environmental impact, easy maintenance, and portability.

Pinnipeds, such as seals and sea lions, have demonstrated remarkable abilities in tracking prey. In a study, the reaction of harbor seals to hydrodynamic stimuli was examined, revealing their capability to detect water velocity as low as 245 μm/s [5]. Even blindfolded, seals could follow the trail of a small submarine with a wake defect velocity of approximately 2 m/s for up to 20 s [6]. Compared with harbor seals, sea lions exhibit lower sensitivity, which is attributed to the unique wave-like shape of the harbor seal’s whiskers [7]. The noise generated by vortex-induced vibrations may mask the weak wake vortex signals of fish. However, the suppression of these vibrations by seal whiskers reduces noise, improves the signal-to-noise ratio, and enhances the detection capabilities of the whiskers. The special shape of seal whiskers, characterized by a wavy surface and an elliptical cross-section, was found to contribute to the suppression of vortex-induced vibrations [8,9]. Morrison et al. [10] conducted a numerical study of the ability of harbor seal whiskers to identify wakes at an Re = 500 using the lattice Boltzmann method (LBM). They revealed that the turbulent kinetic energy of the wake is an order of magnitude lower in the case of a vibrissa-shaped cylinder compared with a circular cylinder. Further research compared the vibrissa-shaped cylinder with cylinders of different shapes, including elliptical cylinders and wave-shaped cylinders, all with the same hydrodynamic diameter. The study utilized particle image velocimetry (PIV) technology at Re = 1800 and observed a considerably reduced recirculation zone in the nodal plane, decreased velocity fluctuation intensities, and the presence of stable reversed flow in the vibrissa-shaped cylinder compared with the other three systems [11]. Furthermore, the vibrissa-shaped cylinder exhibited significant suppression of vortex-induced vibrations even at an Re ≈ 50,000 [12]. Gong et al. [13] examined how artificial disturbances like flapping paddles notably alter the lift forces on harbor seal vibrissae while having little effect on cylinders. Zhao et al. [14] experimentally investigated the vibration responses, spectral frequencies, and fluid forces of the whisker model downstream of a fixed cylinder and a flapping caudal fin at different angles of attack

Over the past decade, significant progress has been made in the development of sensors inspired by sea whiskers. An artificial whisker sensor with a micro-electro-mechanical system was proposed that was able to detect minute disturbances underwater with a velocity detection limit as low as 193 μm/s [15]. An array of whisker-style sensors was designed and used to successfully detect the same hydrodynamic disturbances that triggered changes in the seal’s tracking behavior [16]. Compared with a circular cylinder, the wake of the seal-vibrissa-shaped cylinder was found to present more stable three-dimensional separation, a longer vortex formation length, and a weaker vortex strength in the large eddy simulation framework at an Re = 20,000 [17]. Verma et al. [18] combined numerical simulations and Bayesian experiments to determine the optimal placement of shear and pressure sensors, aiming to identify the location of disturbance sources more effectively. Additionally, Zheng et al. [19] conducted numerical simulations to study the noise generated by nine different shapes of seal-whisker-style cylinders affected by vortex-induced vibrations, aiming to improve biomimetic sensing cylinder models.

To determine the shape and position of obstacles ahead based on the flow signals sensed by seal-whisker-style cylinders, one may need to solve the inverse problem of the three-dimensional fluid–structure problem. This typically exhibits high complexity and nonlinearity, making it difficult and computationally intensive to solve, and often requires the solution of ill-posed problems. However, animals can accurately sense the desired results through their whiskers without complex “calculations”. This has led to the promise of using deep learning models inspired by biological neural networks. Deep learning has now found a wide range of applications in science and engineering, offering exceptional versatility in mapping complex nonlinear relationships without the need for strong prior assumptions about the model structure. Carrillo et al. [20] utilized artificial neural networks (ANNs) and two-dimensional lattice Boltzmann numerical simulation methods to estimate the position and size of obstacles in pipe flow. They used the dynamic pressure (q) or the x-component of velocity ($v_{x})$ profiles to learn obstacle dimensions and locations. Lakkam et al. [21] employed ANNs to determine the shape parameters of complex-shaped hydrofoils in uniform flow, utilizing data from a potential flow model with 25 sensor arrays. Du et al. [22] predicted the Reynolds numbers and cylinder positions using a multi-layer perceptron neural network based on velocity fields calculated by a computational fluid dynamics model. While numerous works have proposed the prediction of location and other information using neural networks based on experimental data, most of these studies focus on artificial lateral lines. Zheng et al. [23] predicted the dipole source positions two body lengths away using a generalized regression neural network with nine sensing points distributed in a cross shape. Wolf et al. [24] used eight two-dimensional sensing points to measure the flow field velocity and predicted the location, distance, and direction of obstacles using an extreme learning machine neural network. Pu et al. [25] explored the localization of dual vibration sources using multi-layer perceptron networks based on air pressure values obtained from sensing points and studied the impact of sensor layout and quantity on localization results. Bodaghi et al. [26] employed a computational fluid–structure interaction model combined with a deep learning approach to decode the intricate mechanisms of seal-whisker sensing, predicting the location and orientation of obstacles based on whisker array signals.

Although there have been many works on obstacle detection sensors that incorporate AI algorithms, most of these studies focus on the recognition capabilities of sensor arrays. We believe that a thorough understanding of the recognition capabilities and underlying mechanisms of individual whiskers is essential for optimizing whisker arrays. However, current research on this topic is still limited. In this paper, we aim to experimentally measure the response and time evolution of lift and drag forces on a single seal-whisker-style cylinder due to excitations caused by upstream obstacles. Additionally, we construct a convolutional neural network (CNN) by Pytorch 1.12.1 to explore the recognition capability of the model.

The paper is organized as follows: Section 2 introduces the experimental setup and procedures, as well as the structure and training settings of the CNN model used in the study. Section 3 presents the CNN model classification results based on the experimental data. Section 4 further analyzes and discusses the results. Finally, the summary and conclusions are provided in Section 5.

## 2. Materials and Methods

### 2.1. Experimental Setup and Procedure

The experiment was conducted in the low-turbulence circulation water channel at the Fluid Mechanics Laboratory of Tianjin University. The testing section of the experimental water channel had a length of 2370 mm, a height of 400 mm, and a width of 306 mm. The upper part of the water channel was open, and the flow velocity of the water in the tank could be adjusted by changing the motor speed. The water depth in the water channel during the experiment was 330 mm. The flow velocity range in the water channel was 0–0.4 m/s, and the turbulence intensity was less than 1% at a flow velocity of 0.4 m/s.

Figure 1 shows the schematic diagram and photo of the experimental platform. The seal-whisker-style cylinder was fixed at the top of the water channel using a support bracket, with the bottom end free, forming a vertically hanging cantilever beam. The top of the seal-whisker-style cylinder was equipped with a dual-axis force sensor (ME-Meßsysteme GmbH, Brandenburg, Germany). The target cylinder to be tested was located upstream of the seal-whisker-style cylinder, directly in front along the flow direction, with the top fixed onto the support bracket and the bottom end free. The immersed length of all the test targets and the seal-whisker-style cylinder was 273 mm. Both the target cylinder and the seal-whisker-style cylinder had their axes perpendicular to the horizontal plane, and the distance (L) between them could be adjusted using the bracket. The dual-axis force sensor installed on the top of the seal-whisker-style cylinder could simultaneously measure the drag force (along the flow direction) and lift force (along the horizontal plane) acting on the seal-whisker-style cylinder. The force signals were output in the form of dual-channel signals and transmitted to a computer through a signal amplifier (Forcechina Co. Ltd., Shanghai, China) and a data acquisition card (National Instruments Corporation, Austin, TX, USA). The signal sampling rate during the experiment was set to 1 KSa/(s·ch).

The seal-whisker-style cylinder used in the experiment was made of resin using 3D printing, as shown in Figure 2a. It was an enlarged version, scaled up 30 times, of the seal-whisker shape parameters proposed by Hanke et al. [8] (Figure 2b). The shape parameters of this seal-whisker-style cylinder included the lengths of the major and minor axes of two controlling elliptical cross-sections (a, b, k, l), the distance between the two controlling elliptical cross-sections (M), and the inclination angle of the two controlling elliptical cross-sections (α, β). The parameters of the spotted seal-whisker-style cylinder used in the experiment were M=27.3 mm, a=17.85 mm, b=7.2 mm, α=15.27°, k=14.25 mm, l=8.7 mm, and β=17.6°. The narrow-face equivalent diameter of the seal-whisker-style cylinder in the Y-Z plane was
(1)$$d=2\times \frac{b+l}{2}=15.90\;mm$$The wide-face equivalent diameter of the seal-whisker-style cylinder in the Y-Z plane was
(2)$$d_{wide}=2\times \frac{a\cos\alpha +k\cos\beta }{2}=30.81\;mm.$$The length of the cylinder was 343 mm.

The nine different shapes of the targets shown in Figure 3 were tested in the experiment, and these targets were all made of resin using 3D printing. The length of all the target models was 375 mm. We used three different diameters of straight cylinders, C1 (15.9 mm), C2 (31.8 mm), and C3 (47.7 mm). By comparing the classification results of the CNN model for these three cylinders, we could test the recognition ability of the seal-whisker-style sensor for test targets of the same shape but different sizes. We also selected two different types of dual-stepped cylinders, both with a larger diameter of 31.8 mm and a smaller diameter of 15.9 mm. The lengths of their larger sections were different, being 35 mm (D1) and 140 mm (D2), respectively. By comparing the recognition results of the CNN model for these two dual-stepped cylinders with those of the straight cylinders C1 and C2, we could test the recognition ability of the seal-whisker-style sensor for these targets. We also used a regular triangular prism with a base side length of 15.9 mm and a square prism with a base side length of 15.9 mm, which equals the diameter of C1. By comparing the recognition results of the CNN model for these two prisms with the straight cylinder C1, we could investigate the recognition ability of the sensor for test targets of the same size but with different shapes.

To test whether the seal-whisker-style sensor can recognize different upstream flow conditions of the same target, we used different orientations for the triangular prism and the square prism in the experiment. For the triangular prism, we used two installation orientations: edge-facing flow (T1) and face-facing flow (T2). For the square prism, we used two installation orientations: face-facing flow (S1) and edge-facing flow (S2). The triangular prisms or square prisms installed in different ways in the experiment were considered as different test targets, so there were a total of nine different test targets.

During the experiment, the seal-whisker-style cylinder and test targets were fixed at the designated distances according to the experimental conditions, and the water flow velocity was adjusted to the required velocity for the conditions. When the incoming flow stabilized, the lift and drag forces acting on the seal-whisker-style cylinder were measured for 122 s. Experiments were conducted for each test target at flow velocities of 0.1 m/s, 0.15 m/s, and 0.2 m/s with the seal-whisker-style cylinder positioned 16 cm away from the test targets. Experiments were also conducted for each test target at a flow velocity of 0.1 m/s with the seal-whisker-style cylinder positioned 16 cm, 24 cm, and 32 cm away from the test targets. The full experimental conditions and the corresponding Reynolds numbers are shown in Table 1. A no-test-target case was also included for flow velocities at 0.1 m/s, 0.15 m/s, and 0.2 m/s. Each test target (including the no-test-target case) was repeated 9 times under each condition to obtain sufficient experimental data for building the sample set used for CNN training and testing. In the absence of the target, the experimental conditions of Case 4 and Case 5 are the same as Case 1, so only the experiment in Case 1 without the target is sufficient.

### 2.2. CNN Structure and Settings

CNN methods initially achieved success in handwritten recognition, and later they were applied to image processing. Currently, CNNs are widely used in deep learning tasks such as electrocardiogram (ECG) signal recognition [27] and bearing-fault diagnosis [28]. CNNs consist of multiple convolutional layers and pooling layers, used for feature extraction and data reduction, followed by fully connected layers and output layers for classification and result output. The advantages of CNN mainly include two points: firstly, CNN can automatically learn features of data, avoiding the limitations of relying on manually selecting features; secondly, CNN has the characteristics of local connections and weight sharing, reducing the number of training weights and the difficulty of network training.

Figure 4 and Table 2 provide the structure and parameters of the CNN we used. Note that the network structure and parameters in Table 2 were chosen by trial and error. The model consists of three convolutional blocks for extracting signal features and three fully connected layers. Each convolutional block consists of a convolutional layer, a batch normalization layer, and a max pooling layer. The first convolutional block uses a one-dimensional convolutional layer with a kernel size of 256 and a stride of 64, which helps the model identify features with longer periods in the signal. The convolutional kernels in the other two convolutional blocks have a size of 7 and a stride of 1. The output of the last convolutional block is transmitted to the three fully connected layers for classification after flattening. Except for the last fully connected layer that outputs the final result using the SoftMax activation function, the other convolutional layers and fully connected layers in the CNN use the Leaky ReLU function as the activation function. The input size of the network is determined by the length of the segments in the sample set and the number of selected channels, and the output result is the likelihood score of various test target categories, which is a one-hot-encoded vector. The category corresponding to the component with the highest probability is the classification result predicted by the CNN.

The force signals from the seal-whisker-style sensor after removing the direct current component are segmented into fragments of different lengths. These signal fragments are then used as the sample set for training and validating the CNN model after undergoing different filtering processes. To study the influence of the length of sample signals on the recognition ability of the CNN model for test objects, we segmented the original signals into fragments with lengths of $2^{12}$, $2^{13}$, $2^{14},$ and $2^{15}$. We selected the fragments with a length of $2^{14}$ and filtered them with cutoff frequencies of 50 Hz, 30 Hz, 10 Hz, and 5 Hz for low-pass filtering and with a cutoff frequency of 5 Hz for high-pass filtering. Sample sets obtained from different filtering methods were used to investigate the influence of signal features in different frequency ranges on the recognition ability of the CNN model. To study the influence of information contained in different channels on classification, we selected unfiltered samples with a length of $2^{14}$, retaining only the lift or drag signals, resulting in two sample sets composed of single-channel signal samples. The specific settings of all the sample sets are listed in Table 3.

To eliminate randomness in the training process of the CNN model, we employed a 3-fold cross-validation method. For each shape, the results of 9 experiments under one condition were divided into 3 folds, with each fold consisting of 3 experiment results. Each training session selected two folds for training and used the remaining fold as the test set to verify the recognition ability of the model for test objects.

The optimizer, loss function, and learning parameters of the proposed CNN are described in Table 4. To prevent overfitting, an early stopping strategy was employed during training. Twenty percent of the training set was randomly selected as the validation set. When the loss function on the validation set did not decrease continuously for 7 epochs, it was considered that the CNN model had converged, and the training was stopped. The training results of the epoch with the minimum loss function were considered to be the final results.

## 3. Results

### 3.1. Validation of the CNN Model

To validate the effectiveness of the CNN model used in this study, we performed fault recognition using drive-end accelerometer signals from the Bearing Fault dataset provided by the Case Western Reserve University Bearing Data Center [29]. We used signals of normal operating conditions, as well as signals of inner race faults (with fault diameters of 0.007″, 0.014″, and 0.021″), outer race faults (with fault diameters of 0.007″, 0.014″, and 0.021″), and ball faults (with fault diameters of 0.007″, 0.014″, and 0.021″) sampled at a frequency of 12 kSa/s; these signals were measured at a 0 HP motor load. Segmentation of the data resulted in a 10-class dataset with segments of length 2048 and a 50% overlap. For training, segments up to 6.67 s were used as the training set, with a random 20% subset reserved for validation, and segments from 6.67 s to 10 s were used for testing. The CNN model achieved classification accuracies of 96.82% on the training set, 94.27% on the validation set, and 93.29% on the test set. These results confirm the model’s ability to identify faults from time-series signals.

### 3.2. Experimental Results of Seal-Whisker-Style Sensor Force Signals

Figure 5 presents the experimental results of lift and drag signals for the targets C1, D2, S1, and for the no-target scenarios. Figure 5a–c correspond to the following test targets: a cylinder with a diameter of 15.9 mm (C1), a dual-stepped cylinder with a thicker segment length of 140 mm (D2), and a cube target with a side length of 15.9 mm (S1). Lift and drag data were recorded under various flow velocities and distances, as shown in Table 1 for the five cases. Figure 5d illustrates the time series of lift and drag forces on the seal-whisker-style cylinder measured at three flow velocities with no upstream target (No). Notably, when no target is present upstream, the lift and drag signals received by the seal whisker style sensor have amplitudes below 0.01 N, displaying high-frequency signals with small amplitudes and no discernible pattern. Conversely, comparing Figure 5a–c with Figure 5d, we find that the presence of an upstream target results in significantly higher lift signal amplitudes and clear quasi-periodic oscillations, while the drag amplitude remains lower. The force signals captured by the model effectively indicate the presence of nearby upstream targets.

For the cylinder with a diameter of 15.9 mm (target C1), positioned at a fixed distance of 16 cm from the seal-whisker-style cylinder, the lift force increases with escalating flow velocity in Cases 1, 2, and 3. Conversely, at a constant flow velocity of 0.1 m/s, as the distance between the target and the seal-whisker-style cylinder extends to 24 cm and 32 cm in Cases 4 and 5, the lift force amplitude experiences a slight decrease while exhibiting heightened high-frequency components in the lift signal. Target D2 displays lift signal characteristics comparable to those of target C1, albeit with a more chaotic signal profile. Similarly, target S1 exhibits overall lift signal traits akin to target C1, but with more pronounced signal periodicity and lower characteristic frequencies. Across the experiments, the drag amplitudes generally remain significantly smaller than the lift amplitudes. However, exceptions arise, such as in Cases 4 and 5 in Figure 5a, where the drag amplitude irregularly surges, occasionally surpassing the lift amplitude. This anomaly is likely attributed to experiment-induced disturbances. While the lift and drag force signals from other sets of experiments mirror the characteristics observed in the C1, D2, and S1 trials, variations in waveform, amplitude, and frequency are evident. The experimental results indicate that force signals for the same test target fluctuate under different flow velocities and distances, while force signals for different test targets diverge even when subjected to the same flow conditions and distances.

### 3.3. Results of the CNN Model

Figure 6 depicts how different sample lengths, filtering methods, and force signal channels affect the CNN model’s accuracy in recognizing test targets using their respective test sets. Noticeable discrepancies in performance between the training and test sets indicate potential issues of severe overfitting and the limited generalization capabilities of the model. As shown in Figure 6, the CNN model trained on sample sets featuring lift channels of length $2^{14}$ or longer without filtering (Sample Sets 3, 4, and 11 from Table 3), as well as sample sets with lift channels filtered using high-pass filters at cutoff frequencies of 50 Hz, 30 Hz, 10 Hz, and 5 Hz (Sample Sets 5, 6, 7, and 8 from Table 3), achieved classification accuracies above 81%, with minimal variance among them. Notably, the model trained on Sample Set 7 (length $2^{14}$ with both lift and drag channels, filtered at a 10 Hz cutoff frequency for high-pass filtering) exhibited the smallest disparity in accuracy between the training and test sets, indicating the least amount of overfitting in comparison with models trained on other sample sets.

From Figure 6, it is evident that as the sample length increases from $2^{12}$ (equivalent to a sample duration of 4.096 s) to $2^{14}$ (corresponding to a sample duration of 16.384 s), the CNN model’s recognition accuracy improves by approximately 16%. However, beyond this point, further increases in sample length result in minimal changes in the model’s recognition accuracy. Moreover, the CNN model trained on sample sets comprised solely of lift signals demonstrates higher recognition accuracy for test targets compared with the CNN model trained on sample sets containing only drag signals.

Next, we present the training and performance details of the CNN model using Sample Set 7 (length $2^{14}$, subjected to 10 Hz low-pass filtering, encompassing both lift and drag signals) as an illustrative example.

Figure 7 presents the evolution of accuracy and cross-entropy loss during the training of the CNN model using Sample Set 7. Initially, there is a rapid decline in cross-entropy loss across both the training and validation sets, accompanied by a swift increase in accuracy. By the 10th epoch, fluctuations in the loss and accuracy stabilize, with the validation set achieving its lowest loss at the 21st epoch. Subsequently, there is no further decrease in loss over the subsequent 7 epochs, leading us to conclude that the CNN model has optimally converged at this stage without displaying significant overfitting tendencies. The accuracy rates for this model on the training, validation, and test sets in the three-fold validation are 95.53 ± 3.26%, 87.69 ± 1.14%, and 84.29 ± 1.16%, respectively.

In the experiment, seven different target shapes from Figure 3 were used as test targets, including triangular prism and square targets placed in various orientations, along with a scenario without any upstream targets, resulting in a total of 10 target categories for CNN classification. Figure 8 illustrates an example using Sample Set 7 to display the distribution of predicted measurements for each test target category. To measure the confusion between two categories, the confusion rate between classes A and B is defined as follows:(3)$$r_{AB}=\frac{N_{A\to B}+N_{B\to A}}{N_{A}+N_{B}},$$
where $N_{A}$ and $N_{B}$ respectively refer to the number of samples with the true label as A or B. $N_{A\to B}$ represents the number of samples with the actual label A but predicted as B, $N_{B\to A}$ denotes the number of samples with the actual label A incorrectly predicted as B, and vice versa.

Figure 8a presents the recognition results on the validation set. It can be observed that in the validation set, the highest confusion rate is between C1 (a straight cylindrical column with a diameter of 15.9 mm) and T1 (a regular triangular prism with an edge length of 15.9 mm facing upstream), reaching 36.7%. S1 (a square column with a side length of 15.9 mm facing sideways) and T2 (a regular triangular prism with a side length of 15.9 mm facing sideways) follow, with a confusion rate of 10.7%. D1 (a dual-stepped cylindrical column with a coarse segment length of 35 mm) and D2 (a dual-stepped cylindrical column with a coarse segment length of 140 mm) also exhibit some confusion, with a confusion rate of 6.8%, whereas errors in predictions for the remaining categories are minimal. In the validation set, there is significant confusion between C1 and T1, with a confusion rate exceeding 35%, while relatively minor confusion exists between S1 and T2, as well as between D1 and D2.

From Figure 8b, it can be observed that in the test set, 268 samples actually belonging to C1 were identified as T1, and 251 samples actually belonging to T1 were classified as C1, resulting in a confusion rate $r_{T1,C1}=33.3\%$; 111 samples actually belonging to S1 were identified as T2, and 181 samples actually belonging to T2 were classified as S1, giving a confusion rate of 18.7%. Additionally, 47 samples actually belonging to D1 were identified as D2, and 98 samples actually belonging to D2 were classified as D1, resulting in a confusion rate of 9.3%. Furthermore, 37 samples actually belonging to C2 were identified as S2, and 48 samples actually belonging to S2 were classified as C2, representing a confusion rate of 5.4%. Moreover, there were instances where 33 samples actually belonging to C2 were identified as D2, and 26 samples actually belonging to C2 were classified as T2, while errors in predictions for the remaining categories were minimal. In the test set, there was significant confusion between C1 and T1 and between S1 and T2, with confusion rates exceeding 18%, and there was also some confusion between D1 and D2, albeit to a lesser extent.

Figure 9 presents the CNN classification results under varying experimental conditions in the form of a confusion matrix, with the actual target categories on the horizontal axis and the predicted target categories on the vertical axis. Notably, there was notable confusion between C1 and T1, as well as between S1 and T2, across all experimental settings. However, in experiments with a distance of 16 cm and flow velocities of 0.15 m/s (Case 2) and 0.2 m/s (Case 3), the confusion between S1 and T2 was relatively less pronounced compared with other scenarios. At a distance of 16 cm and a flow velocity of 0.1 m/s (Case 1), some incorrect predictions were observed, including some misclassifications such as C2 samples being misidentified as D2, S1, S2, and T2, and certain S2 samples being incorrectly labeled as C2, D2, and T2. In experiments with a flow velocity of 0.1 m/s and distances of 16 cm (Case 1), 24 cm (Case 4), and 32 cm (Case 5), there was also subtle confusion between D1 and D2.

## 4. Discussion

### 4.1. The Impact of Vortex Shedding Frequency on Force Signals

Periodic shedding vortices occur when a steady flow passes over a bluff body, and it is influenced by the flow velocity U and the object’s shape and size. The vortex shedding frequency of a circular cylinder f can be calculated as [30]:(4)$$f=St\frac{U}{D},$$
where U is the inflow fluid velocity, D is the diameter of the circular cylinder, and St is the Strouhal number. For Reynolds numbers Re=UD/ν (where ν is the kinematic viscosity of water) between 1000 to 10,000, the Strouhal number is approximately equal to 0.2. In our experiments, the Reynolds number varied from 1600 to 9600 under the current setup.

Figure 10 displays the experimental vortex shedding frequency data for the three straight circular cylinders (C1, C2, and C3 in Table 1) used as test targets, alongside the theoretical shedding frequencies calculated from Equation (4). The measured downstream force signal frequencies represent the average of the peak frequencies from nine measurements. From Figure 10, it is evident that the force signal frequency of the seal-whisker-style sensor is closely linked to the vortex shedding frequency of the upstream cylinders. This observation suggests that the seal-whisker-style sensor effectively captures the disruption signals emitted by upstream objects, highlighting the prominence of vortex characteristics as essential criteria for the sensor in identifying upstream objects.

### 4.2. The Influence of Sample Length and Filtering on the CNN Model

As previously mentioned, an increase in sample length leads to improved model performance; however, this enhancement reaches a limit, and once the sample length reaches $2^{12}$, further improvement in the model’s performance is not observed. This limitation may be attributed to the vortex fields generated by the upstream cylinders, which contain information of varying periods. Extending the sample length can expose more characteristics of long-period information, thereby improving the CNN classification outcomes for test targets. Nevertheless, the useful information contained within the sample has its limitations, and beyond a certain length, the additional extension does not yield more useful information, therefore plateauing the accuracy improvement of the CNN model.

Figure 11 demonstrates the influence of sample length on the CNN model performance. It is apparent that with a sample length of $2^{12}$ (Sample Set 1), there is significant confusion between C1 and T1, between D1 and D2, and between S1 and T2, as well as some minor confusion between C2 and C3, between C2 and S2, and between S1 and S2. As the sample length increases to $2^{14}$ (Sample Set 3) and $2^{15}$ (Sample Set 4), the level of confusion between C1 and T1, S1 and T2, and D1 and D2 decreases compared with that for the sample length of $2^{12}$, although it remains noticeable. Meanwhile, confusion between the other test targets is minimal. While increasing the sample length enhances the CNN model’s performance for the test targets, it does not eliminate confusion between categories like C1 and T1, D1 and D2, and S1 and T2.

As depicted in Figure 6, CNN models trained on sample sets filtered with low-pass filters at cutoff frequencies of 50 Hz, 30 Hz, 10 Hz, and 5 Hz exhibited similar recognition accuracy on the test set compared to models trained on unfiltered sample sets, achieving recognition accuracies of around 83.1%. These results indicate that the essential components of the force signal predominantly operate below 5 Hz, consistent with the shedding frequency range of the target. Moreover, this suggests that our CNN model can effectively mitigate high-frequency noise interference, showcasing its robustness. Conversely, a high-pass filtered sample set at a cutoff frequency of 5 Hz resulted in significantly lower recognition accuracy of only 31.7 ± 4.9% on the test set, highlighting once more the importance of the force signal’s components operating below 5 Hz.

### 4.3. Lift Signal Spectra and Their Impact on Target Recognition

Based on the previous discussions, when utilizing the trained CNN model to identify targets, confusion is evident between C1 and T1, between T2 and S1, and between D1 and D2. Despite adjustments to the sample lengths and to the signal filtering, these confusions persist. Hence, we postulate that these uncertainties may stem from the inherent characteristics of the signal samples.

Notably, from Figure 6, it is observed that a CNN model exclusively trained on unfiltered lift signals achieved an accuracy of 81.0 ± 2.3% on the test set. Similarly, a CNN model trained on both lift and drag signals of equivalent lengths (Sample Set 3) exhibited an accuracy of 83.1 ± 1.49% on the test set, showcasing minimal disparity between them. Conversely, a CNN model trained solely on drag signals yielded an accuracy of only 54.8 ± 4.0% on the test set, indicating a limited recognition capability. Consequently, it can be deduced that the primary basis for the CNN model’s identification of target cylinders lies within the characteristics of the lift signals, as drag signals provide less relevant feature information. Hence, the analysis of signal features predominantly focuses on lift signals.

By performing Fourier transform on the lift signals in Sample Set 10, their amplitude spectra are obtained. Figure 12 illustrates the highest peak values along with their corresponding frequencies in the resulting amplitude spectra, where closely clustered points suggest similarities in their primary peak characteristics. Specifically, Figure 12a reveals similarities in the highest peak values and corresponding frequencies of lift signals for C1 and T1 across all experimental conditions. Moreover, as depicted in Figure 9, the CNN model demonstrates confusion in distinguishing between these two categories under various experimental scenarios. Similarly, at a flow velocity of 0.1 m/s (Cases 1, 4, and 5), there is a strong similarity between the signals of T2 and S1, resulting in severe confusion between them. However, as the flow velocity surpasses 0.1 m/s, although the peak frequencies of the wake signals for T2 and S1 remain close, disparities in the peak heights result in reduced confusion compared with that for the 0.1 m/s flow velocity.

In a 1D CNN architecture, convolutional kernels serve as filters for distinct waveforms, with each kernel aligned to a unique waveform type. The convolution process involves calculating the inner product of the signal segment with the kernel waveform to gauge the similarity between them, akin to the short-time Fourier transform (STFT). While the function of convolution kernels in the CNN resembles triangular basis functions in the STFT, they are consistently optimized during the neural network training phase. In 1D CNN models directly processing time-series data, the convolution layer aims to extract frequency domain features from the input signal. The more similar the frequency components present in the input signal, the higher the similarity in the output results for the CNN model. Consequently, signals sharing similar frequency domain features, such as the aforementioned C1 and T1 and the T2 and S1 wake signal samples, are more prone to confusion.

## 5. Conclusions

We conducted an underwater detection experiment and integrated it with a CNN model to investigate the recognition capability and mechanism of a single seal-whisker-style sensor on flow signals. The experiment involved measuring force signals from the seal-whisker-style cylinder in nine tested targets under five different working conditions, which were then utilized for training and testing using the CNN model.

The CNN model test results revealed that, in most instances, the model could successfully identify various targets, although there were instances of misidentification. Increasing the sample length moderately improved the recognition effectiveness; however, further extension beyond a certain point did not enhance performance. Upon comparing the CNN model’s performance trained using differently filtered sample sets, it was observed that components with frequencies exceeding 5 Hz provided little useful information. Lift signals exhibited more distinctive features for differentiation, while drag signals displayed less significant characteristics. The primary basis for the seal-whisker-style sensor in distinguishing between different target cylinders lies in the lift signal.

When testing a cylinder as the upstream target, the average frequency of the sensor’s lift signal closely matched the vortex shedding frequency of the cylinder, indicating that important and relevant information was carried by the upstream vortices. Analyses of the lift signal’s frequency spectrum in correlation with the confusion matrix unveiled that the model’s identification efficacy across various targets relied heavily on disparities in the frequency spectrum characteristics of the lift signal.

Our research findings suggest that integrating a single seal-whisker-style sensor with a CNN enables the identification of underwater objects based on the model’s responsiveness to inflow, particularly the vortices within it. Nonetheless, the CNN model tends to struggle with distinguishing signals exhibiting similar frequency domain features, necessitating further enhancements for its recognition performance. For future work, introducing spatial distribution information of signals through the formation of sensor arrays featuring multiple seal-whisker-style sensors promises to provide richer insights into the CNN model’s identification process.

## Figures and Tables

**Figure 1 sensors-24-05418-f001:**
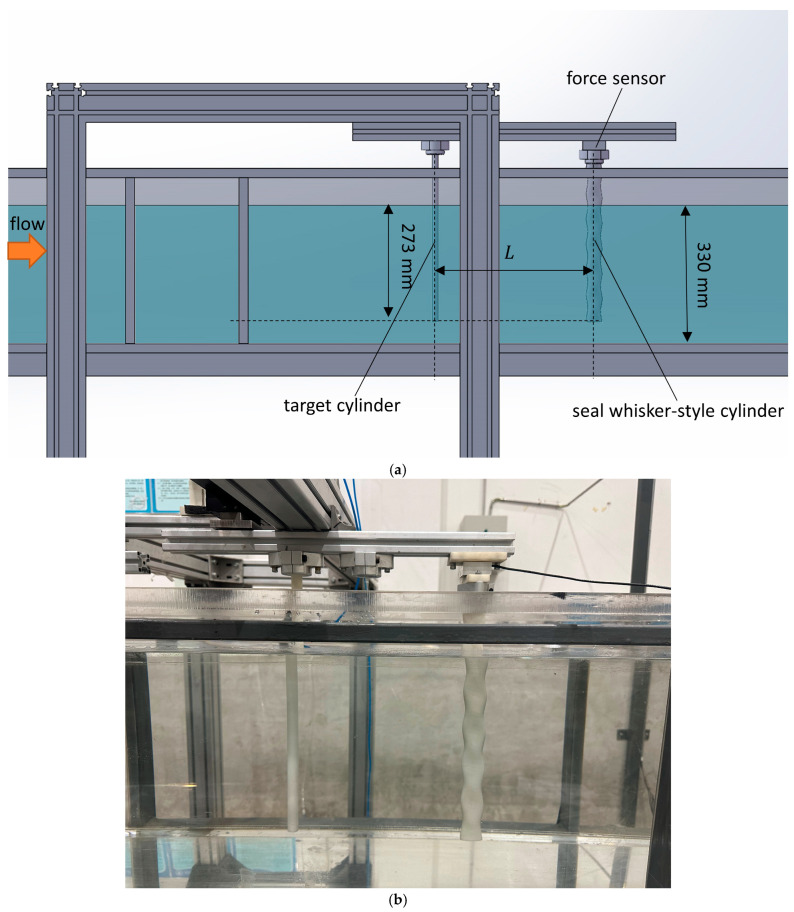
Experimental setup. (**a**) schematic diagram, (**b**) photograph.

**Figure 2 sensors-24-05418-f002:**
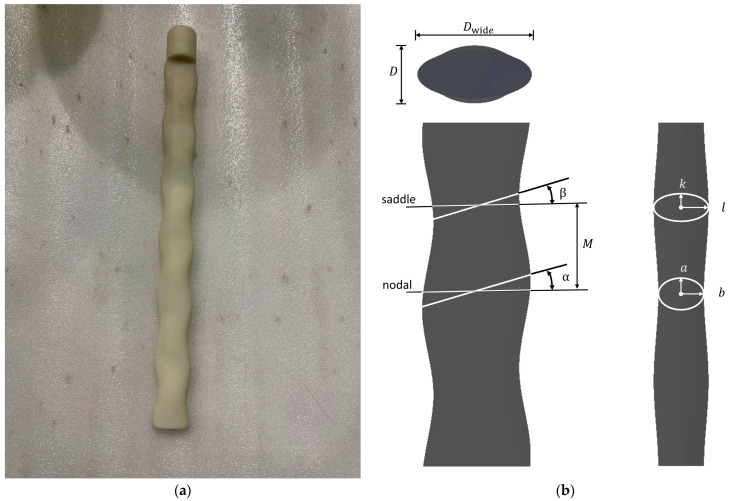
(**a**) Photo and (**b**) schematic diagram of the seal-whisker-style cylinder.

**Figure 3 sensors-24-05418-f003:**
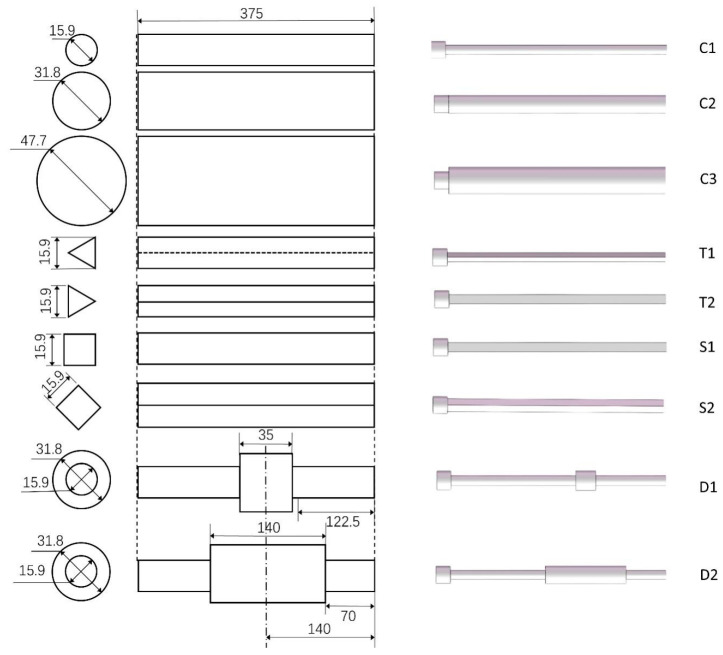
Shapes and dimensions of the tested targets. (**Left**) cross-section; (**Middle**) side view; (**Right**) 3D view. Units: mm.

**Figure 4 sensors-24-05418-f004:**
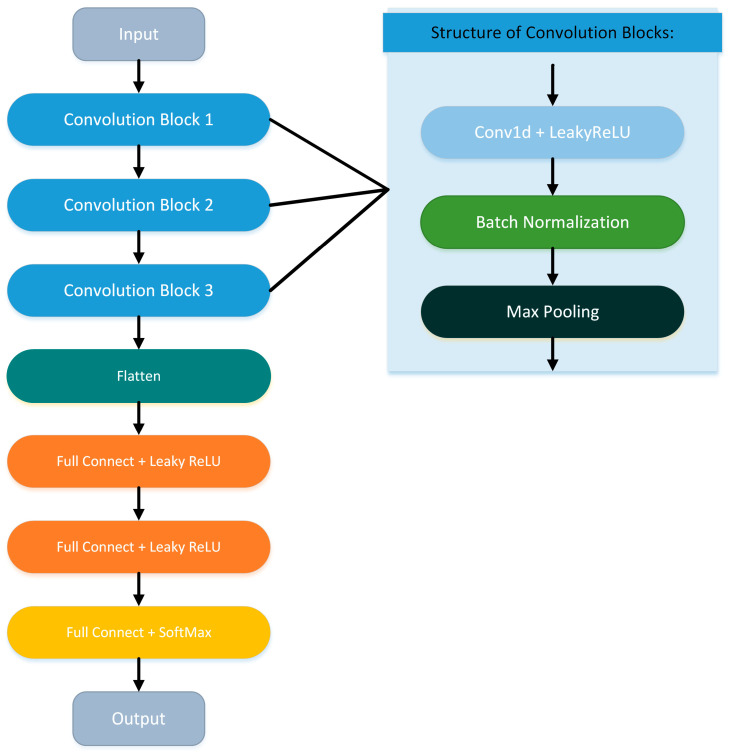
CNN structure.

**Figure 5 sensors-24-05418-f005:**
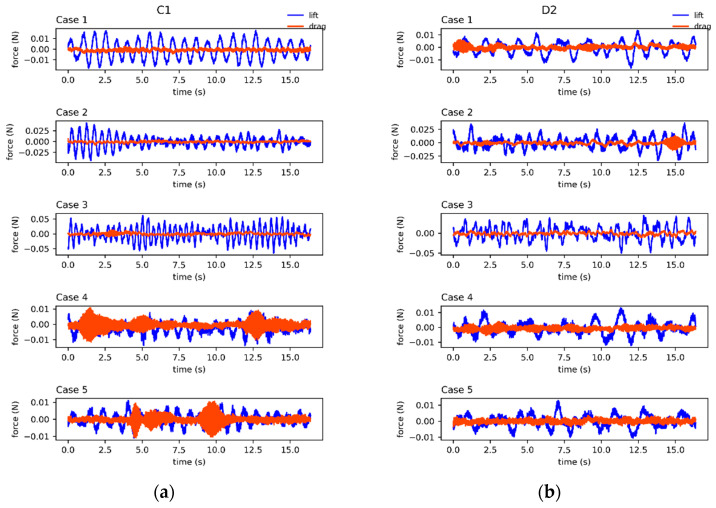
The force signals acting on the seal-whisker-style cylinder under experimental conditions for the targets (**a**) C1 (cylinder with a diameter of 15.9 mm), (**b**) D2 (dual-stepped cylinder with a thicker segment length of 140 mm), and (**c**) S1 (cube with a side length of 15.9 mm), and (**d**) with no target (No), with a time duration 16.384 s.

**Figure 6 sensors-24-05418-f006:**
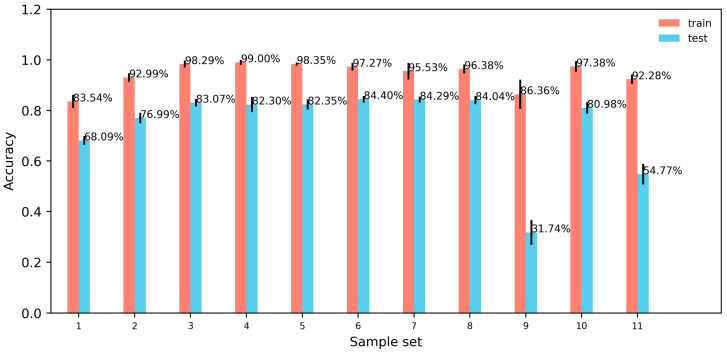
The recognition accuracy of test targets by the CNN model trained on 11 different sample sets.

**Figure 7 sensors-24-05418-f007:**
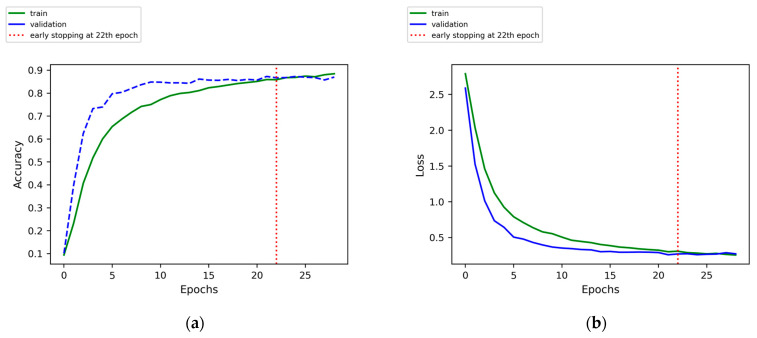
The training progress curve of accuracy and cross-entropy loss for the CNN model trained using Sample Set 7. (**a**) the training accuracy history of CNN. (**b**) the training loss history of CNN.

**Figure 8 sensors-24-05418-f008:**
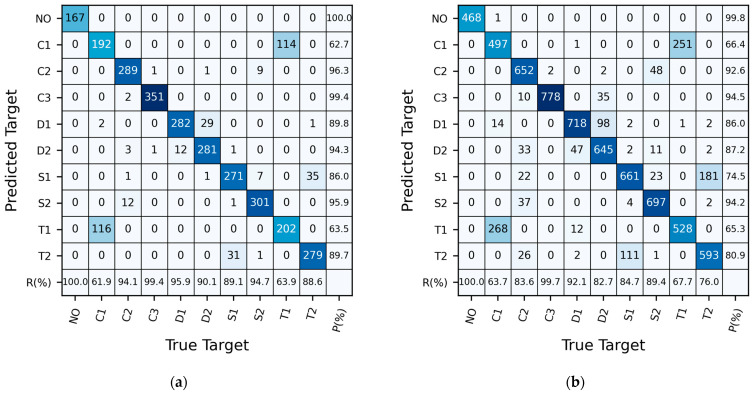
The confusion matrix of the CNN model trained on Sample Set 7. (**a**) The validation set; (**b**) The test set. Here, P denotes the precision rate, and R denotes the recall rate.

**Figure 9 sensors-24-05418-f009:**
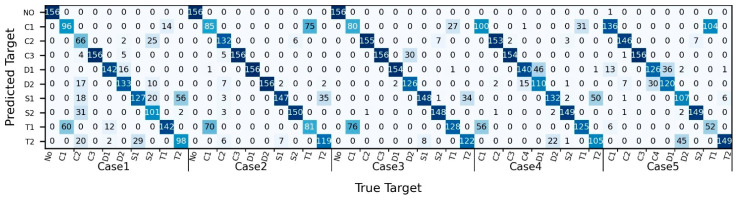
Detailed results of the test targets trained on Sample Set 7.

**Figure 10 sensors-24-05418-f010:**
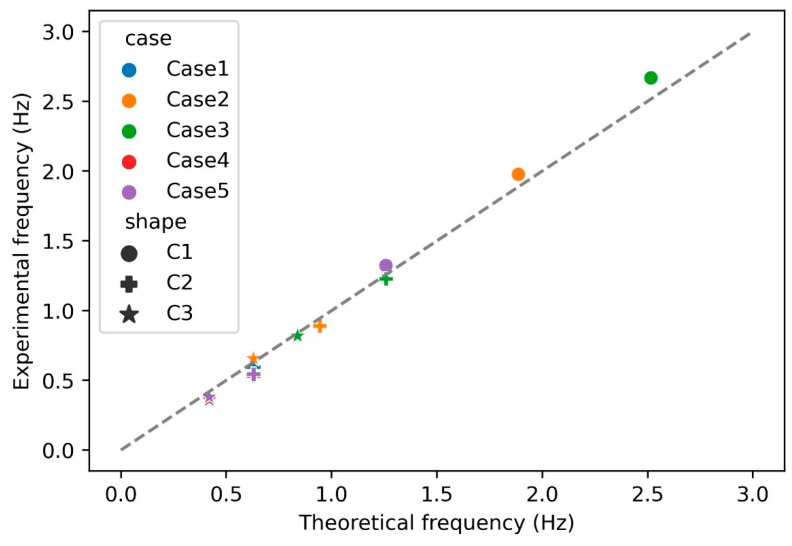
The theoretical vortex shedding frequency versus experimental results for cylinders C1, C2, and C3 from Table 1.

**Figure 11 sensors-24-05418-f011:**
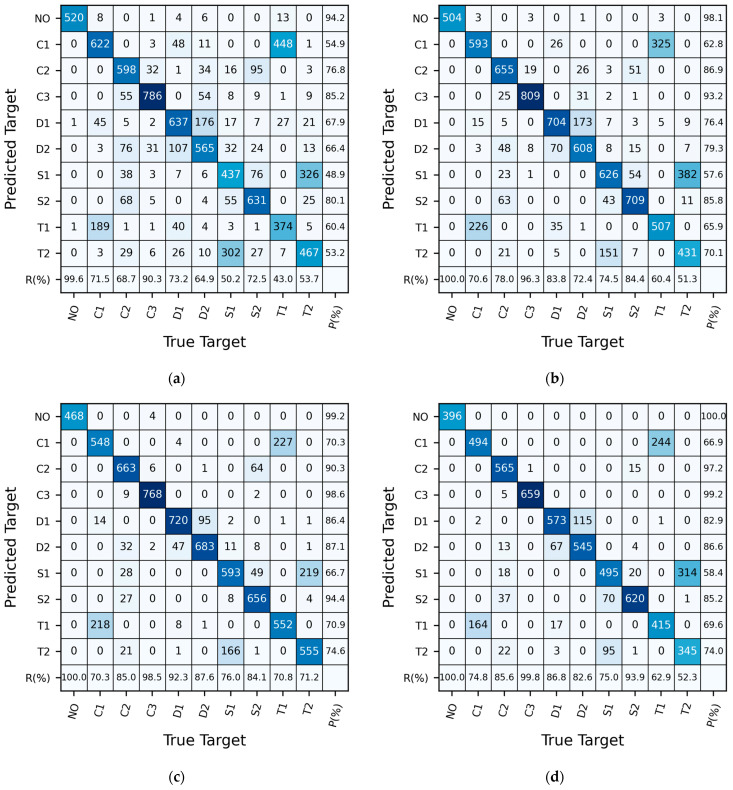
The confusion matrices for the test sets of CNN models trained on sample sets with different lengths. (**a**) Sample Set 1, length $2^{12}$; (**b**) Sample Set 2, length $2^{13}$; (**c**) Sample Set 3, length $2^{14}$; (**d**) Sample Set 4, length $2^{15}$ (sample sets as shown in Table 3).

**Figure 12 sensors-24-05418-f012:**
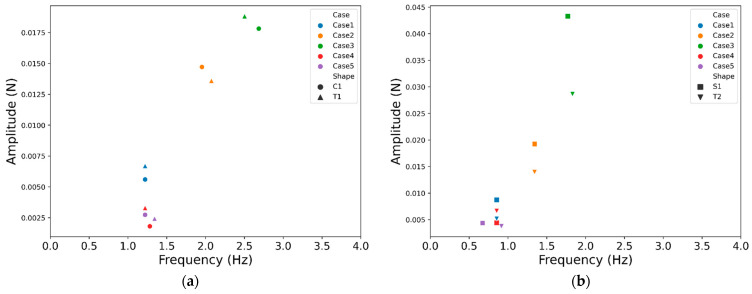
The peak frequency spectrum of the lift signals in Sample Set 10 (**a**) C1 and T1; (**b**) S1 and T2.

**Table 1 sensors-24-05418-t001:** Experimental conditions, showing the incoming flow velocity and the distance from the upstream axis of the test target to the axis of the seal-whisker-style cylinder. “No” in the table indicates no test target. (Cases marked with “√” was conducted, while cases marked with “-” were deemed unnecessary to conduct).

	C1	C2	C3	T1	T2	S1	S2	D1	D2	No
Case 1 U=0.1 m/s, L=16 cm	√	√	√	√	√	√	√	√	√	√
Case 2 U=0.15 m/s, L=16 cm	√	√	√	√	√	√	√	√	√	√
Case 3 U=0.2 m/s, L=16 cm	√	√	√	√	√	√	√	√	√	√
Case 4 U=0.1 m/s, L=24 cm	√	√	√	√	√	√	√	√	√	-
Case 5 U=0.1 m/s, L=32 cm	√	√	√	√	√	√	√	√	√	-

**Table 2 sensors-24-05418-t002:** CNN parameters.

Network Structure		Parameters
Convolution Block 1	Conv1d + Leaky ReLU	filter size: 256; stride: 64; channel: 64; padding: no padding; negative slope: 0.01
	Batch Normalization	-
	Max Pooling	pooling size: 4; stride: 1; padding: no padding
Convolution Block 2	Conv1d + Leaky ReLU	filter size: 7; stride: 1; channel: 64; padding: no padding; negative slope: 0.01
	Batch Normalization	-
	Max Pooling	pooling size: 4; stride: 1; padding: no padding
Convolution Block 3	Conv1d + Leaky ReLU	filter size: 7; stride: 1; channel: 64; padding: no padding; negative slope: 0.01
	Batch Normalization	-
	Max Pooling	pooling size: 4; stride: 1; padding: no padding
Flatten		-
Full Connect + Leaky ReLU		node number: 120; dropout: 0.2; negative slope: 0.01
Full Connect + Leaky ReLU		node number: 80; dropout: 0.2; negative slope: 0.01
Full Connect + Softmax		node number = number of classes

**Table 3 sensors-24-05418-t003:** The training and testing sample sets of the CNN.

Sample Sets	Length	Time Step	Filtering	Channel	Train Set	Validation Set	Test Set
Sample Set 1	$2^{12}$	2048	unfiltered	lift and drag	13,363	3341	8352
Sample Set 2	$2^{13}$	2048	unfiltered	lift and drag	12,902	3226	8064
Sample Set 3	$2^{14}$	2048	unfiltered	lift and drag	11,981	2995	7488
Sample Set 4	$2^{15}$	2048	unfiltered	lift and drag	10,138	2534	6336
Sample Set 5	$2^{14}$	2048	50 Hz low-pass	lift and drag	11,981	2995	7488
Sample Set 6	$2^{14}$	2048	30 Hz low-pass	lift and drag	11,981	2995	7488
Sample Set 7	$2^{14}$	2048	10 Hz low-pass	lift and drag	11,981	2995	7488
Sample Set 8	$2^{14}$	2048	5 Hz low-pass	lift and drag	11,981	2995	7488
Sample Set 9	$2^{14}$	2048	5 Hz high-pass	lift and drag	11,981	2995	7488
Sample Set 10	$2^{14}$	2048	unfiltered	lift only	11,981	2995	7488
Sample Set 11	$2^{14}$	2048	unfiltered	drag only	11,981	2995	7488

**Table 4 sensors-24-05418-t004:** Learning settings.

Learning Settings	
Optimizer	Adam
Loss function	Categorical cross-entropy function
Learning rate	0.0001
Batch size	32
Maximum number of training epochs	75
Patience of early stopping	7

## Data Availability

Dataset available on request from the authors.

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
