# Peer review of "Shape Classification Using a Single Seal-Whisker-Style Sensor Based on the Neural Network Method"

_sensors, 2024, doi:10.3390/s24165418_

Round 1

Reviewer 1 Report

Comments and Suggestions for Authors

1. The spelling and grammar of the entire manuscript need to be further revised

2. The formulas in the manuscript need to be further checked for correctness and the explanation of the formulas is reasonable

3. The pictures shown in the manuscript need further description, and it is recommended to redraw Figure 1 and Figure 2

4. It is recommended to unify the numbering of pictures and tables, as the numbering of figures and tables is repeated throughout the manuscript

5. The manuscript recommends further verifying the effectiveness of the proposed method on other data sets

6. It is recommended to unify the format of the references in the manuscript

Comments on the Quality of English Language

The spelling and grammar of the entire manuscript need to be further revised

Author Response

Thank you very much for taking the time to review this manuscript. Please find the detailed responses below and the corresponding revisions highlighted changes in the re-submitted files.

Comments 1: The spelling and grammar of the entire manuscript need to be further revised

Response 1: Thank you for pointing this out. We have now critically gone through the manuscript and have fixed the grammatical errors in the revised manuscript.

Comments 2: The formulas in the manuscript need to be further checked for correctness and the explanation of the formulas is reasonable

Response 2: Agreed. We have thoroughly checked all the formulas and provided additional descriptions where necessary.

Comments 3: The pictures shown in the manuscript need further description, and it is recommended to redraw Figure 1 and Figure 2

Response 3: Thank you for pointing this out. Figure 1 and Figure 2 have been redrawn with additional details.

Comments 4: It is recommended to unify the numbering of pictures and tables, as the numbering of figures and tables is repeated throughout the manuscript

Response 4: Thank you for the suggestion. We have revised the manuscript to ensure consistent numbering of figures and tables, eliminating any repetitions.

 Comments 5: The manuscript recommends further verifying the effectiveness of the proposed method on other data sets

Response 5: Agreed. Section 3.1 is added to verify the effectiveness of our CNN model on the Bearing Fault dataset provided by Case Western Reserve University.

Comments 6: It is recommended to unify the format of the references in the manuscript

Response 6: Thank you for the suggestion. We have revised the manuscript to ensure that the format of the references is consistent throughout.

Reviewer 2 Report

Comments and Suggestions for Authors

Commets for authors:

1. The aim of the article is to recognize the data obtained from the seal whisker style sensor using the CNN network and evaluate the importance of recognizing lift and drag signals. But when the literature was reviewed, the work done on the recognition of lift and drag signs was almost not considered.

2. Although it was stated that a single sensor is ineffective for object detection in many previous works, the authors used a single sensor in their research. The importance of using a single sensor rather than a multiple sensor is not explained by the authors.

3. The authors did not provide a comparative analysis of the advantages of the seal whisker style sensor compared to the important researches carried out for object detection using AI algorithms, the scope of novelty obtained in this research was not clearly seen.

4. Out of 26 articles in the list of references, 10 articles were written 10 years ago, 9 articles were written 5 years ago. Seven of the new literatures were written after 2019. In order to improve the quality of the article, it is necessary to study the work done after 2020 and add it to the article.

5. In Figure 4, the CNN structure is not drawn on demand, and the names of the layers should be redrawn/written correctly, for example, the letters in the convolution, flatten, and output layers are dropped. Also, in Figures 4 and 7 (which should actually be Figures 8 and 11), the recall value in the first column of the confusion matrix is ​​not quite clear, for example, it's harder to guess if it's 100% or 0%. The images should be revised as required by MDPI and presented with the confusion matrix values ​​visible.

6. The authors mismatched in ordering the figures, for example, after figure 4, the figures were again numbered as figure 1. The order of explaining the figures in the text is correct. However, the sequence of correspondence between the descriptions in the text and the figures is confused. Therefore, the authors should revise this procedure and correct the deficiencies.

7. Although data is given about the size of the dataset developed after pre-processing for CNN training, but how many sample sets are there in total, how many of them are the training set and how many are the validation set, and only the test set information can be seen from the confusion matrix. It is necessary to provide information about this and fulfil the gaps

8. In order to use abbreviations in the article, it is necessary to provide an extended view first, for example, MEMS, DC is not given at the text.

Author Response

Thank you very much for taking the time to review this manuscript. Please find the detailed responses below and the corresponding revisions highlighted changes in the re-submitted files.

Comments 1: The aim of the article is to recognize the data obtained from the seal whisker style sensor using the CNN network and evaluate the importance of recognizing lift and drag signals. But when the literature was reviewed, the work done on the recognition of lift and drag signs was almost not considered.

Response 1: Thank you for pointing this out. However, Existing literature on lift and drag forces primarily focuses on studying the seal whisker’s own response to lift and drag, as well as comparisons with other cylindrical shapes like round cylinders. In terms of obstacle detection using sensors, studies such as references 19 and 25 utilize pressure signals, while references 20-24 rely on velocity field measurements. To date, we have not encountered any research that uses lift and drag forces as sensor signals in a deep learning network to recognize upstream obstacles. Our work aims to utilize CNNs to analyze lift and drag signals obtained from a seal whisker-inspired sensor and to assess the effectiveness of these signals for obstacle recognition.

Next, we will provide response to comments 2 and 3 taken together.

Comments 2: Although it was stated that a single sensor is ineffective for object detection in many previous works, the authors used a single sensor in their research. The importance of using a single sensor rather than a multiple sensor is not explained by the authors.

Comments 3: The authors did not provide a comparative analysis of the advantages of the seal whisker style sensor compared to the important researches carried out for object detection using AI algorithms, the scope of novelty obtained in this research was not clearly.

Response 2&3: Although there has been many works on obstacle detection sensors that incorporate AI algorithms, most of these studies focus on the recognition capabilities of sensor arrays. We believe that a thorough understanding of the recognition capabilities and underlying mechanisms of individual whiskers is essential for optimizing whisker arrays. However, current research on this topic is still limited.

Comments 4: Out of 26 articles in the list of references, 10 articles were written 10 years ago, 9 articles were written 5 years ago. Seven of the new literatures were written after 2019. In order to improve the quality of the article, it is necessary to study the work done after 2020 and add it to the article.

Response 4: Thank you for pointing this out. Reference 13, 14 and 26 have been added to the revised manuscript.

Comments 5: In Figure 4, the CNN structure is not drawn on demand, and the names of the layers should be redrawn/written correctly, for example, the letters in the convolution, flatten, and output layers are dropped. Also, in Figures 4 and 7 (which should actually be Figures 8 and 11), the recall value in the first column of the confusion matrix is ​​not quite clear, for example, it's harder to guess if it's 100% or 0%. The images should be revised as required by MDPI and presented with the confusion matrix values ​​visible.

Response 5: Agree. Figure 4 , Figure 8, and Figure 11 have been redrawn for clarity.

Comments 6: The authors mismatched in ordering the figures, for example, after figure 4, the figures were again numbered as figure 1. The order of explaining the figures in the text is correct. However, the sequence of correspondence between the descriptions in the text and the figures is confused. Therefore, the authors should revise this procedure and correct the deficiencies.

Response 6: We apologize for the mistake. We have revised the manuscript to ensure consistent numbering of figures and tables, eliminating any repetitions.

Comments 7: Although data is given about the size of the dataset developed after pre-processing for CNN training, but how many sample sets are there in total, how many of them are the training set and how many are the validation set, and only the test set information can be seen from the confusion matrix. It is necessary to provide information about this and fulfil the gaps

 Response 7: Thank you for pointing this out. Further details have been given in table 3 in the revised manuscript.

Comments 8: In order to use abbreviations in the article, it is necessary to provide an extended view first, for example, MEMS, DC is not given at the text.

Response 8: Thank you for pointing this out. These have been fixed in the revised manuscript.

Round 2

Reviewer 2 Report

Comments and Suggestions for Authors

According to the "Response 2&3: Although there has been many works on obstacle detection sensors that incorporate AI algorithms, most of these studies focus on the recognition capabilities of sensor arrays. We believe that a thorough understanding of the recognition capabilities and underlying mechanisms of individual whiskers is essential for optimizing whisker arrays. However, current research on this topic is still limited" authors should add the importance of the work to the article.

Author Response

Thank you very much for taking the time to review this manuscript. Please find the detailed responses below and the corresponding revisions highlighted changes in the re-submitted files.

Comment 1: According to the "Response 2&3: Although there has been many works on obstacle detection sensors that incorporate AI algorithms, most of these studies focus on the recognition capabilities of sensor arrays. We believe that a thorough understanding of the recognition capabilities and underlying mechanisms of individual whiskers is essential for optimizing whisker arrays. However, current research on this topic is still limited" authors should add the importance of the work to the article.

Response 1: The reviewer is completely correct and our modifications reflect the suggestions.